# Advances in Genomics and Postgenomics in Poultry Science: Current Achievements and Future Directions

**DOI:** 10.3390/ijms26178285

**Published:** 2025-08-26

**Authors:** Irina Gilyazova, Gulnaz Korytina, Olga Kochetova, Olga Savelieva, Elena Mikhaylova, Zilya Vershinina, Anna Chumakova, Vitaliy Markelov, Gulshat Abdeeva, Alexandra Karunas, Elza Khusnutdinova, Oleg Gusev

**Affiliations:** 1Institute of Biochemistry and Genetics–Subdivision of the Ufa Federal Research Centre of the Russian Academy of Sciences (IBG UFRC RAS), 450054 Ufa, Russia; gilyasova_irina@mail.ru (I.G.); guly_kory@mail.ru (G.K.); olga_mk78@mail.ru (O.K.); olyasavelie@yandex.ru (O.S.); mikhele@list.ru (E.M.); zilyaver@mail.ru (Z.V.); nettachak@yandex.ru (A.C.); marckelow.vitalick2017@yandex.ru (V.M.); gulshatik2001@mail.ru (G.A.); elzakh@mail.ru (E.K.); 2Life Improvement by Future Technologies (LIFT) Center, 121205 Moscow, Russia; 3Intractable Disease Research Center, Graduate School of Medicine, Juntendo University, Tokyo 113-8421, Japan

**Keywords:** poultry industry, *Gallus gallus domesticus*, whole-genome and transcriptome studies, quantitative trait locus (QTL), poultry reproduction, molting, metagenomic research, infectious disease resistance, environmental stress adaptation, CRISPR/Cas genomic editing, multi-omics approaches

## Abstract

The poultry industry, a globally fast growing agricultural sector, provides affordable animal protein due to high efficiency. *Gallus gallus domesticus* are the most common domestic birds. Hybrid chicken breeds (crosses) are widely used to achieve high productivity. Maintaining industry competitiveness requires constant genetic selection of parent stock to improve performance traits. Genetic studies, which are essential in modern breeding programs, help identify genome variants linked to economically important traits and preserve population health. Next-generation sequencing (NGS) has identified millions of single nucleotide polymorphisms (SNPs) and insertions/deletions (INDELs), enabling detection of genome-wide regions associated with selection traits. Recent studies have pinpointed such regions using broiler lines, laying hen lines, or pooled genomic data. This review discusses advances in chicken genomic and transcriptomic research focused on traits enhancing meat breed performance and reproductive abilities. Special attention is given to transcriptome studies revealing regulatory mechanisms and key signaling pathways involved in artificial molting, as well as metagenome studies investigating resistance to infectious diseases and climate adaptation. Finally, a dedicated section highlights CRISPR/Cas genomic editing techniques for targeted genome modification in chicken genomics.

## 1. Introduction

The poultry industry is one of the fastest-growing agricultural sectors globally. It remains the most affordable source of animal protein due to its high productivity and relatively low production costs. *Gallus gallus domesticus* is the most common domestic bird. More than 80 million tons of eggs and approximately 65 billion poultry are produced worldwide annually to meet the global demand for meat in the commercial sector [1].

To achieve high productivity results, crosses—hybrids of chicken breeds obtained by crossbreeding—are used. Maintaining the efficiency and competitiveness of the poultry industry requires constant selection to improve the performance of the parent stock. Breeding and pedigree-based studies in modern poultry farming are fundamentally dependent on genetics. Genetic studies allow identifying genome variants associated with important economically valuable traits and maintaining the genetic health of the population. Research aimed at obtaining highly productive chicken crosses using modern genetic technologies, such as high-throughput analysis of the whole genome, is highly relevant. The chicken genome has served as an essential resource for the global research community since 2004, when the first draft genome of the wild ancestor of the domestic chicken was published [2].

The chicken is currently an important model organism in the fields of medicine and embryology, immunogenetics, oncology, and nutrigenomics. Despite its relatively small genome size, the chicken genome contains almost the same number of genes as the human genome. Due to its in ovo development and easy access to and manipulation of the chicken embryo using incubated eggs, the chicken occupies an intermediate position between humans and lower vertebrates. Indeed, chickens serve as an important model organism for the study of vertebrate developmental biology and germline development in both vertebrates and birds. Furthermore, chickens are also an important model for genetic and genomic studies due to their ease of breeding and rearing as well as their long history of selective breeding, which has resulted in considerable phenotypic and genetic diversity [3].

Domestic chickens have hundreds of different breeds, but commercial chickens can be divided into two main groups, broilers (meat type) and laying hens (egg type), both of which have been subjected to artificial selection for centuries. These two types have different phenotypic and genotypic characteristics. Broiler breeding focuses on growth and musculature development, such as body weight, feed conversion, and breast weight. In contrast, the breeding of laying hens focuses mainly on reproductive traits, such as egg production and egg quality. Millions of SNPs and INDELs from different chicken genomes have been identified using next-generation sequencing (NGS) data, and now it is possible to detect genome-wide regions of selection traits. Recent studies in chickens have identified regions susceptible to selection using only broiler lines, only laying hen lines, or pooled sample datasets [4].

In this review, we aimed to discuss recent advances in genomic and transcriptomic studies of chickens for traits associated with improved performance of meat chicken breeds and reproductive abilities of poultry. The review strategy involved searching for publications and information in the GWAS Atlas https://ngdc.cncb.ac.cn/gwas (accessed on 20 May 2025), NCBI https://www.ncbi.nlm.nih.gov (accessed on 20 May 2025), Ensembl https://www.ensembl.org/info/about/publications.html (accessed on 20 May 2025), and ChickenGTEx Atlas https://ngdc.cncb.ac.cn/chickengtex (accessed on 20 May 2025) databases, primarily from 2014 to 2025, using keywords related to genomic, transcriptomic, and metagenomic studies of domestic chickens and CRISPR/Cas genomic editing. Special attention is paid to transcriptome studies to identify regulatory mechanisms and key signaling pathways associated with artificial molting. Additionally, we examine metagenome studies investigating chicken resistance to infectious diseases and environmental stressors such as climate conditions. Finally, we have provided a separate section that highlights the CRISPR/Cas genomic editing techniques in chicken genomics (Figure 1).

## 2. Whole-Genome and Transcriptome Studies of Association with Economically Important Traits in Chickens

### 2.1. Genome-Wide Association Studies with Quantitative Traits of Increased Productivity of Meat Chicken Breed

Quantitative traits such as feed intake (FI), feed conversion ratio (FC), feed utilization efficiency (FE), body weight gain (WG), growth rate and relative growth rate (RG and RGR), body weight indices (BW), residual feed intake (RFI), and feed conversion ratio (FCR) are the most important economic traits in poultry production [5].

The quantitative trait locus (QTL) database for chickens [6] contains 75 quantitative trait loci (QTL) associated with feeding traits, including feed conversion ratio, feed efficiency, feed intake, and residual feed intake [7]. Numerous studies have identified QTLs, candidate genes, and gene variants associated with productive traits in chickens [8,9]. A recent study by Moreira et al. on a Brazilian Embrapa F2 chicken population using a high-density SNPs chip (600 K, Affymetrix^®^, Santa Clara, CA, USA) annotated SNPs in QTL regions of genes related to osteogenesis, skeletal muscle development, growth, energy metabolism, and lipid metabolism. These SNPs may be associated with chicken body weight, feed efficiency, and growth rate, which are breeding targets for improving chicken productivity [10]. In a study by Mebratie et al. potential loci and candidate genes associated with body weight and feeding efficiency were identified in a commercial broiler line. 11 QTLs and 21 SNPs were associated with body weight traits, while 5 QTLs and 5 SNPs were associated with feed efficiency traits [5]. Genotyping a broad panel of 600 K Affymetrix^®^ (Santa Clara, CA, USA) SNPs across 23 Italian chicken populations yielded data on the association of a number of SNPs with traits of adaptation to changing environmental conditions [11]. A genome-wide association study by Cha et al. on Korean domestic chicken samples identified genes and SNPs associated with body weight: *WDR37*, *KCNIP4*, *SLIT2*, *PPARGC1A*, *MYOCD*, and *ADGRA3* genes. The main biological role of the identified candidate genes was found to regulate cellular growth and development, which may be related to the body weight trait [12]. QTLs for the feeding utilization efficiency (FE) trait have been identified and reported on 24 out of 39 chicken chromosomes (GGA), including the sex chromosome Z [13]. The Yang et al. research group performed an RNA-seq analysis of the pectoral muscle obtained from chickens with extreme opposite scores of the residual feed intake trait (RFI) and revealed that *ND2* (*p_FDR_* = 0.012), *ND4* (*p_FDR_* = 0.010), *CYTB* (*p_FDR_* = 0.007), *RAC2* (*p_FDR_* = 0.012), *VCAM1* (*p_FDR_* = 0.0003), *CTSS* (*p_FDR_* = 0.0001), and *TLR4* (*p_FDR_* = 0.014) were key genes for the efficiency of the feed of local chickens. These genes may influence feed efficiency through deep involvement in reactive oxygen species (ROS) production and inflammatory response [14].

The whole genome sequencing of ten traditional Chinese chicken breeds with yellow plumage has revealed that the key genes associated with meat quality include ryanodine receptor (genes *RYR2* and *RYR3*), *IL-18*, *F-box* protein FBXO5, collagens *COL1A2*, *COL4A2*, *COL6A1*, *COL6A2*, *COL4A1*, and *COL23A1*, myostatin *GDF8*, a member of the *HSPA5* heat shock protein family, and type I transmembrane protein *SHISA9*. Notably, the *RYR* gene plays an important role in the development of skeletal muscle and the formation of pale, soft, and juicy meat in domestic chicken, and *COL1A2* is an important gene influencing meat quality in chickens. The estimation was made using the π- ratio test [15].

Excess fat deposition is a negative factor in poultry meat production because it reduces feed utilization efficiency, increases the cost of meat production, and poses a health risk to consumers. A genome-wide association study (GWAS) analysis performed on chickens from the Brazilian population revealed 22 unique QTL variants associated with abdominal and carcass fat content, located in the region of 26 candidate positional genes. These genes were involved in biological processes such as fat cell differentiation, insulin and triglyceride levels, and lipid biosynthesis. The heritability of these traits has been shown to range from 0.43 to 0.56 [16].

Several genes involved in lipid metabolism and the development of adipose tissue in chickens (*ATPR2*, *APOB*, *PPARG*, *ZNF423*, *IGFBP2*, *ADCY2*, *AKAP6*, *SCARB1*, and *PLA2R1*) have also been identified in a number of studies involving selection signature analysis [17,18]. A detailed catalog of genetic variants, both lineage-specific and common to the two Brazilian broiler and laying hen lines, is described. The catalog can be used for future genomic studies involving association analysis with relevant phenotypes, therefore facilitating the identification of causative mutations in chickens. Furthermore, it provides a foundation for Marker Assisted Selection (MAS) or genomic selection for important traits in chickens [4].

### 2.2. Genome-Wide Association Studies with Traits Related to Poultry Reproduction

The key parameters characterizing the reproductive ability of poultry are duration of fertility, age of puberty in hens, and egg production, which includes a whole range of quantitative characteristics such as total egg count, age of the first egg, weight of the first egg, egg weight, weekly number of eggs, etc. Duration of fertility and age of sexual maturity in hens are important traits in poultry production, directly affecting the productivity and economic efficiency of breeding. Therefore, the investigation of molecular mechanisms that regulate the duration of the fertility trait can enhance our understanding of this trait and improve economic efficiency and animal welfare in hatching egg production. Polymorphisms associated with fertility duration in GWAS have provided valuable insights into the genetic architecture underlying these reproductive traits. Advances in the use of new testing and recording technologies [19] and genetic studies [20] provide promising opportunities to improve fertility duration and reproductive efficiency in commercial chicken breeds.

#### 2.2.1. Duration of Fertility

The GWAS study on Jinghong and Jingfen chickens identified 27 SNPs located in three genomic regions (GGA1: 41 Kb, GGA3: 39 Kb, and GGA8: 39 Kb) significantly associated with the duration of fertility [21]. In a study conducted on Dongxian blue and white leghorn hens, SNPs located in genes involved in hormone regulation, such as *PRLHR*, were found to influence the age of the first appearance of the eggs through oxytocin secretion [22]. In another study, polymorphisms in the *GDF9* gene were found to be associated with both the age of the first egg and egg weight in blue-footed Dongxiang Lukedanji hens (Dongxiang Lukedanji) (*p* < 0.05) [23]. Several key genes and pathways involved in fertility traits have been identified as a result of GWAS studies. For instance, overexpression of the *CYP21A1* gene, involved in steroid hormone biosynthesis, was revealed in high-yielding eggs from White Leghorn hens [24]. A study by Sun et al. identified the zygote arrest 1 (*ZAR1*) gene as a candidate gene associated with the development of the yolk and follicle. ZAR1 protein is involved in lipid transport and lipoprotein synthesis, processes essential for yolk formation and follicle development [25].

#### 2.2.2. Age of Puberty

In poultry, the onset of sexual maturity is usually accompanied by the development of secondary sexual characteristics and reproductive capacity. Puberty in chickens is a multifaceted process influenced by genetic, hormonal, and environmental factors. Research has identified key biomarkers and genetic pathways that play an important role during this stage of development. In particular, circulating microRNAs (miRNAs) and the *KISS-1* gene have been shown to significantly regulate hormonal changes associated with the onset of puberty in chicks [26,27]. To date, there are no unique and highly sensitive biomarkers for measuring the onset of puberty in chicken (*Gallus gallus*). Taking into account the fact that circulating miRNA have been shown to be promising biomarkers for the diagnosis of various diseases and traits, Han et al. performed high-throughput sequencing and analysis of specific miRNA expression profiles in serum and blood plasma of chickens at two different pubertal stages—before puberty (BO) and after puberty (AO). They found that seven miRNAs, including miR-29c, miR-375, miR-215, miR-217, miR-19b, miR-133a, and let-7a, had great potential to serve as novel biomarkers for measuring puberty onset in chickens (*p* < 0.05) [28].

#### 2.2.3. Egg-Laying Capacity

Egg production is one of the most important economic traits in poultry breeding. In 2022, the world chicken egg production is approximately 88.68 million tons, which is about 93% of poultry egg production. The main attributes of egg production include number of eggs, age of first egg, weight of first egg, and average egg weight. Egg number is the most important indicator of production efficiency among these traits, which is directly related to egg production. In addition, improving individual egg production traits is a major goal for many chicken breeding programs. QTL variant mapping and GWAS analysis have been widely used to detect many molecular markers of complex traits in hens, including those associated with egg production traits. A total of more than 540 QTLs associated with egg number, more than 50 QTLs for age of first egg, and more than 400 QTLs for egg weight in hens https://www.animalgenome.org/cgi-bin/QTLdb/GG/index (accessed on 20 May 2025) have been reported in the QTL database [29].

Studies of candidate genes associated with egg production traits often focus on investigating the hypothalamic-pituitary-gonadal endocrine axis [30]. Fedorova et al. found that the heritability coefficient of egg weight in chickens of the Pushkin breed for mothers was 0.701 and for fathers 0.389 [31]. A number of GWAS studies of egg production traits in hens using the 60 K SNP array have been published [32,33,34], or a high-density 600 K SNP chip [23,35,36,37,38] was performed by a study using the genotyping by genome reduction and sequencing (GGRS) approach [22]. GWAS analysis of yellow Jinghai chickens revealed that polymorphisms associated with reproductive traits in chickens are located near the genes *FAM184*, *TTL*, *RGS1*, *FBLN5*, and *PCNX* (*p* < 1.12 × 10^−6^) [33]. Zhao et al. conducted GWAS of commercial egg-laying chicken breeds (White Leghorn, Rhode Island, and Dwarf hen) and local Chinese chicken breeds with low egg production to identify the molecular mechanisms underlying the dramatic differences in egg number between the breeds analyzed. They found 148 polymorphisms associated with egg number traits (*p* < 7.80 × 10^−6^), of which 4 overlapped with previously described quantitative trait loci (QTL) associated with egg production and reproductive traits [38]. SNPs in genes *UPP1* (A79958113G, *p* = 1.73 × 10^−8^), *FHOD3* (C83033381T, *p* = 1.13 × 10^−8^), and *NCALD* (A128890264G, *p* = 7.55 × 10^−10^) were associated with egg weight, and genes *GAL* (G16754641A, *p* = 5.74 × 10^−8^), *CENPF* (G21147253A, *p*-value = 3.84 × 10^−9^), *GPC2* (C3321921T, *p* = 3.42 × 10^−8^), *PEMT* (C5246144G, *p*-value = 9.00 × 10^−8^), *CYP3A4* (T4287370A, *p*-value = 4.84 × 10^−8^), and *TMEM200B* (G2854459T, *p*-value = 6.53 × 10^−8^) with body weight at first oviposition (have been identified in GWAS of chickens from China [39]. In resequencing of 287 female yellow broiler chicken breeds in China (Qingyuan hen), the metastasis-associated protein 2 (*MTA2*) gene was found to be a functional gene influencing egg production [40]. The use of the whole-genome sequencing approach compared with SNP chips allows us to identify all genetic variants and significantly increases the accuracy and power of genetic analysis to detect SNPs associated with complex traits. RNA sequencing of ovarian tissues from F1 chickens obtained from crosses between Chinese indigenous spotted chickens and Chinese indigenous yellow chickens with pendulous-comb, PC) and upright-comb, differential expression of genes predominantly involved in biological processes related to proteins, lipids, and nucleic acids, including protein biosynthesis and lipid transport, has been identified. Hens with drooping combs have been shown to have higher egg production and higher levels of reproductive hormones compared to hens with vertical combs, supporting the theory that the hen comb phenotypes are closely related to the reproductive capacity of hens. A comprehensive analysis of whole genome sequencing and RNA sequencing data from the study samples identified 30 potential candidate genes associated with egg production in hens (*CAMK1D*, *CLSTN2*, *MAST2*, *PIK3C2G*, *TBC1D1*, *STK3*, *ADGRB3*, and *PPARGC1A*) (*p* < 0.05) [41]. Egg laying is a sensitive period for any bird species. The egg laying rate is equal to the number of eggs laid divided by the number of days in the record period. This trait has the highest economic weight in laying hen breeding programs. The duration and number of clutches are controlled by the ovulatory cycle, directly observed by the oviposition cycle and circadian rhythm, as well as the internal cycle of follicle growth and maturation. GWAS analysis of Laiwu Black breed (indigenous Chinese breed) hens revealed 421 polymorphisms significantly associated with oviposition traits [42].

#### 2.2.4. Quality and Integrity of Eggshells

In the global egg industry, the integrity of the eggshell is critical to the economic viability of poultry production. The shell must be strong enough to withstand mechanical stress during packaging and transport. The eggshell also forms an embryonic chamber for the developing chick, providing mechanical protection, allowing gas exchange, and preventing microbial contamination. Chicken eggshells are a bioceramic material composed of columnar crystals of calcite (CaCO_3_) and an organic protein matrix. The size and orientation of calcium carbonate crystals affect the quality of the shell [43]. GWAS analysis of hens obtained from reciprocal crosses between White Leghorn (WL) and Dongxiang (DX) showed that the inheritance coefficient of eggshell weight ranged from 0.30 to 0.46, eggshell thickness from 0.21 to 0.31, and eggshell strength from 0.20 to 0.27. The most important gene variants associated with eggshell quality (eggshell weight, eggshell thickness and eggshell strength) include SNPs in inositol 1, 4, 5-trisphosphate receptor type 2 (*ITPR2*) gene (rs316607577 and rs316447591), involved in the regulation of intracellular calcium ion transport in the uterus and promotes eggshell calcification, the *PIK3C2G* gene (rs312347405), acting as calcium-dependent phospholipid binding motifs, and non-SMC condensin I complex subunit G (*NCAPG*, rs14491030), which has been shown to influence eggshell quality in young hens (*p* < 9.08 × 10^−7^) [44]. Zhang et al. found 889 differentially expressed genes in utero between Rhode Island White hens with low eggshell strength and normal eggshell strength. Many genes were involved in calcification-related processes, including calcium ion transport and calcium signaling pathways [45]. GWAS of chickens from the F2 population derived from crosses between the White Leghorn breed and the indigenous Dongxiang breed of China revealed genes involved in the regulation of calcium ion concentration in the cytoplasm. These are polymorphisms in phospholipase C Zeta 1 (*PLCZ1*, rs314759160, *p* = 4.96 × 10^−10^), ATP-binding cassette, C9 subfamily member (*ABCC9*, rs314985144, *p* = 2.75 × 10^−8^), inositol 1,4,5-trisphosphate receptor type 2 (*ITPR2*, rs314403945, *p* = 9.18 × 10^−9^), potassium channel internal rectification subfamily J member 8 (*KCNJ8*, rs15301807, *p* = 3.52 × 10^−7^), calcium channel alpha 1 subunit C (*CACNA1C*, rs315771606, *p* = 5.13 × 10^−7^), and islet amyloid polypeptide (*IAPP*, rs312653027, *p* = 5.42 × 10^−9^) genes, associated with regulation of total integrated intensity. Total integrated intensity in the analysis of chicken eggshells is a quantification of X-ray fluorescence used to determine the composition of mineral and organic components in the shell, which allows us to estimate the content of elements such as calcium, magnesium, phosphorus, and other minerals that influence shell strength and quality [43]. In the GWAS study of Rhode Island Red hens, two genomic regions on chromosome 1 were found to be associated with eggshell strength. The *FRY* gene (gga1p176000693, *p* = 1.17 × 10^−6^, gga1p175972447, *p* = 1.82 × 10^−6^), which encodes a microtubule-binding protein and plays an important role in various cell skeleton assemblies, the pecanex homologue 2 gene (*PCNX2*, gga3p39043602, *p* = 4.10 × 10^−6^), and the ENSGALG00000052468 gene (gga1p23249882, *p* = 2.57 × 10^−6^) are supposed to be the main candidates associated with eggshell strength [46].

To systematize information, we used the database https://ngdc.cncb.ac.cn/gwas (accessed on 20 May 2025), which includes the results of the GWAS of the breed *Gallus gallus*-5.0. There are 130 traits represented in the database. The loci with the highest level of significance and the traits responsible for high productivity are summarized in Table 1 and Appendix A.

Methodological limits of the existing articles include small sample sizes in a number of articles and differences in allele frequencies due to heterogeneity in genetic data on chicken breeds. Available GWAS databases provide limited information on domestic chickens; trait analysis is predominantly based on the Gallus gallus breed, while other breeds are covered fragmentarily.

## 3. Identification of Regulatory Mechanisms Associated with Artificial Molting of Chickens Using Transcriptome Analysis

Molting is a natural adaptation to climate change in all birds, including chickens. Artificial (forced) molting can rejuvenate and restore the reproductive potential of old hens. In laying hens, the economic benefits of late egg production decline as egg numbers decrease [55]. Artificial molting can trigger a new egg laying cycle, restore high productivity, and improve egg quality and immune function [56]. The molting process usually takes 14–16 weeks, and under natural conditions the egg production of hens gradually decreases. In contrast, artificial molting usually requires only 4 weeks. To reduce molting time and increase economic efficiency, laying hens are often subjected to artificial molting. Artificial molting is a protocol that artificially creates stressors for hens that cause unique morphological and physiological alterations in the reproductive system, resulting in hens that stop laying and lose feathers and weight within a short period of time. Following this phase, hens regrow new feathers and enter a second peak of egg production, continuing to lay for approximately another year [57]. Artificial molting can be induced by various methods, including feed withdrawal or the administration of a high-zinc diet [58]. Among these, feed withdrawal remains the most popular among farmers because of its simplicity [59]. Laying hens are subjected to severe stress, which causes a deficiency of cellular calcium, the main substance supporting neurohormone secretion, and calcium deficiency affects the regulatory function of the hypothalamus [60]. To adapt to starvation stress, the hypothalamic-ovarian-gonadal system regulates lipid metabolism in the liver and maintains stable blood glucose levels at the expense of thyroid or sex hormone production [61]. However, calcium deficiency suppresses the secretion of luteinizing hormone (LH) by the pituitary gland, leading to a decrease in plasma LH levels or even a cessation of estrogen secretion. As a result, hens experience decreased weight and immunity [62], the isthmus of the oviduct degenerates [63], and ovarian aging and atrophy result in reduced egg production and poor eggshell quality. When stress is alleviated by the resumption of water supply, hens meet the body’s energy needs by restoring nervous and endocrine functions [64]. During this recovery phase, the viability of laying hens is gradually restored, with the most obvious manifestations being the regrowth of new feathers, reduced mortality rates, improved eggshell quality [65], and the gradual recovery of egg mass to body weight ratio (higher than before molting) [66].

Additionally, litter bacteria count [67] and bone strength [68] tend to increase. Currently, to increase productivity, fasting is increasingly used; however, the exact molecular mechanism of this process remains unclear. The development of next-generation sequencing technologies allows us to identify differentially expressed genes associated with important economic traits in chickens: growth and development [60,69,70], reproductive traits [14,71,72], and disease resistance traits [73,74]. Nevertheless, investigation of the transcriptome during intermittent starvation in chickens has only been conducted in single studies to date. Zhang et al. performed transcriptome analysis to reveal the dynamic mechanism of regulatory gene expression during artificial molting in laying hens. Blood hormone levels and gene expression levels in the hypothalamus and ovaries of chickens at five different molting periods were measured to identify the genetic mechanism of rejuvenation. Three key hormones, estrogen, growth hormone, and thyroid hormone, were found to be involved in the regulation of molting. Using transcriptome analysis, the authors found that the trends of gene expression levels in the hypothalamus and ovaries at five different stages of molting are generally similar. Among the identified genes, 45 genes regulate cell senescence during starvation, and 12 genes promote cell development during the recovery period in the hypothalamus. Furthermore, five key genes (*INO80D*, *HELZ*, *AGO4*, *ROCK2*, and *RFX7*), associated with senescence, were identified using weighted gene co-expression network analysis (WGCNA). These findings suggest that artificial molting can restore the reproductive function of old chickens by regulating the expression levels of aging- and development-related genes [58]. In the study of Wang et al. 90 laying hens aged 71 weeks with an egg production rate of 60% and comparable body weight were used to explain the mechanism of ovarian function changes during artificial starvation-induced molting. The biological samples (serum and ovarian tissue) were collected on different days of fasting and resumption of feeding. Also, the levels of reproductive hormones in serum and antioxidants in ovarian tissue and the expression levels of genes *KIT*-*PI3K*-*PTEN*-*AKT* and *GDF-9* in ovaries were measured by quantitative real-time PCR at different stages. The results demonstrated that artificial molting activates the KIT-PI3K-PTEN-PTEN-AKT signaling pathway and promotes the activation of primordial follicles during starvation and early resumption of feeding; gonadotropin secretion gradually increases, which promotes the rapid development of primary and secondary follicles to mature follicles and ovulation. This study provides a theoretical basis for improving the ovarian function in laying hens and optimizing the molting program [75]. A recent transcriptomic study of the mechanism underlying induced molting on reproductive function, productivity, and egg quality in a total of 240 Jingfen No. 6 laying hens, which were 380 days old by Ma et al. showed that starvation resulted in atrophy of oviducts, disappearance of large yellow follicles, and decreased serum reproductive hormone levels compared with the pre-molt stage. Transcriptome analysis revealed that the differentially expressed genes were significantly associated with the interaction of cytokines and their receptors, cell adhesion molecules, and the arachidonic acid metabolism signaling pathway during oviduct and ovarian tissue remodeling. The study showed that the key candidate genes were *BMPR1B*, *NEGR1*, *VTN*, and *CHAD*, which play a key role in reproductive remodeling in molted hens [76]. The study by Zhang L used an RNA sequencing approach to elucidate the mechanism of feather and hair follicle growth during forced molting by starvation during the late oviposition stage. Feather and hair follicle samples were collected for hematoxylin and eosin staining, analysis of hormonal changes and follicle characteristics, and for transcriptome sequencing. Molting was observed in chicks from day 13 to day 25 of life and in newborns from day 3 to day 32. During molting, the content of triiodothyronine and tetraiodothyronine increased significantly. After plumage change, calcium content was significantly higher and ash content was significantly lower. Characterization of hair follicles revealed a trend of increased pore density and decreased pore diameter after resumption of feeding. According to the results of RNA sequencing, several major genes, including *DSP*, *CDH1*, *PKP1*, and *PPCKB* genes, were identified, which may be associated with hair follicle growth and development [77].

Thus, current transcriptomic studies in chickens explored various aspects of molting-induced physiological changes. Gene-phenotype correlation analysis and RNA sequencing, signaling pathways, and key genes that may influence changes in various traits in chickens have been identified, but so far the data are largely contradictory due to differences in sample collection, variability in bird age, breed, and housing conditions, as well as different methodologies used, which require standardization for further research.

## 4. Metagenomic Research in Poultry Production

The gut microbiome is a complex and dynamic ecosystem with multiple metabolic and immune functions that are critical to chicken health and productivity [78,79,80,81,82]. The gastrointestinal tract of chickens hosts a wide range of microorganisms that provide hydrolases for birds to ferment a variety of complex carbohydrates [83,84]. Accumulating evidence indicates that the chicken gut microbiome plays a vital role not only in nutrient digestion and absorption [85], but also in immune system development [86], pathogen suppression [87], abdominal fat gain [82], and feed utilization efficiency [88,89,90,91]. Understanding the role of the microbiome is critical for the use of probiotics and prebiotics to enhance immunity and productivity of chickens [92,93,94,95,96,97]. There is evidence of the significant impact of the chicken microbiome on egg quality and safety, both through the vertical transmission of infectious agents via the ‘gut-ovary-egg’ pathway and indirectly, through metabolites of the intestinal microbiota, such as short-chain fatty acids, butyric acid, and tryptophan derivatives, involved in regulating egg quality through the ‘microbiome-intestine-liver/brain-reproductive tract’ system [98]. However, the relationship between intestinal microbiota and egg quality has not yet been sufficiently studied, and further comprehensive research on microbial metabolites affecting egg taste and eggshell quality is required [99,100].

Over the past decade, NGS technology has enabled significant progress in the study of taxonomic and functional diversity within the microbiomes of poultry and other animals [55,80,101,102,103]. For example, 469 microbial genomes were assembled based on metagenomic data (MAG) from blind outgrowths of chick colon [104], and one year later, after analyzing 5595 MAGs, 853 genomes were already assembled [85,105]. The database was further expanded. Metagenomic studies have shown that the intestinal microbiome of chickens is dominated by four bacterial phyla: *Firmicutes*, *Bacteroidetes*, *Actinobacteria*, and *Proteobacteria*. The blind passages of the large intestine of chickens contain the highest number of microbes, and this is where the main fermentation of fiber to form short-chain fatty acids takes place [106]. Carbohydrate-active enzymes (CAZymes, https://www.cazy.org/), which are the major enzymes that break down plant fiber and degrade host-derived dietary carbohydrates and glycans, have received the most attention in studies of gut microbiome studies across both chickens and other animals [107]. For example, more than 8000 CAZymes have been identified from 155 genomes assembled by metagenomic analysis (MAG) from the chicken gut microbiome [85]. Among these, glycoside hydrolases (GH) and glycosyltransferases (GT) were found to be the most abundant functional proteins involved in carbohydrate metabolism, accounting for 45.23% and 37.34% of the total CAZymes, respectively. Therefore, chicken microbiome is a promising source of enzymes and microbial resources for biotechnological applications based on plant biomass fermentation [97]. In addition to its involvement in nutrient metabolism, the chicken gut microbiome is considered as a springboard for antimicrobial resistance genes (ARGs), potentially jeopardizing human health through the widespread use of antibiotics on poultry farms [108,109,110,111]. Mobile genetic elements such as plasmids facilitate the horizontal transfer of antimicrobial resistance genes among bacterial populations [102,106,112,113]. The composition of the gut microbiome is known to change with the age of the host [114,115,116]. Studies based on human and mouse data have shown that the aging process has a strong impact on the distribution of taxonomic groups and functional capacity of the GI bacterial community [114,116,117]. There are a number of papers describing changes in microbiome composition as birds grow older [102,118,119]. As these changes are linked to health status [120,121], there is a need to understand the causes of changes in microbiome composition and mediated gut functionality in order to develop technologies that improve chicken performance and health [85]. Artificial molting techniques increase egg production, improve egg quality, extend the productive lifespan of hens, support healthy offspring, and contribute to feed savings on large farms [75]. Fasting-induced molting (FIM) is a common method used to improve egg production of older laying hens. However, this approach can induce various stresses in hens, such as disturbances in microbiome composition and intestinal inflammation [67,68,122]. It was previously shown that probiotic and vitamin supplementation administered during artificial molting improved the diversity and composition of the microbiome, preventing starvation-induced problems and enhancing immunity, reproductive hormone secretion, and lipid metabolism in the liver after molting [75]. The extent of gut damage, microbiome composition, and metabolome composition were also investigated to elucidate the effects of the FIM process on chicken health. Findings showed that starvation resulted in a marked decrease in villus height and villus/crypt ratio, combined with increased levels of inflammation and intestinal permeability. The composition of the microbiota changed during the fasting period: *Escherichia-Shigella* increased, while *Ruminococcaceae-Lactobacillus* decreased significantly. *Escherichia-Shigella* counts are known to be positively correlated with citrinin and sterobilin, which lead to intestinal inflammation, while *Ruminococcaceae-Lactobacillus* show a positive correlation with lanthionine and reduced glutathione, thereby reducing intestinal inflammation. Therefore, intestinal probiotics containing *Ruminococcaceae* and *Lactobacillus* can be used as potential supplements to enhance gut immune function and fasting-induced intestinal stress [123]. Determining the factors influencing the development of the gastrointestinal microbiota of chickens remains a complex task due to a variety of additional parameters such as age, nutrition, housing conditions, etc. The degree to which these factors influence the colonization of the GI tract of laying hens is currently poorly understood, but with the introduction of various proteomic, transcriptomic, and metabolomic approaches, it is now possible to comprehensively understand the functionality of the gut microbiota, predict possible metabolic pathways, and develop new strategies to improve poultry health and performance. The integration of metagenomic data from the gut microbiome of chickens will complement reference databases and create a catalogue for further characterization of the functional characteristics of different microorganisms and selection of optimal technological solutions, including for safe and efficient artificial molting of laying hens.

## 5. Genetic Studies of Resistance to Infectious Diseases in Chickens

Resistance to infectious diseases in chickens is a polygenic trait that includes various genes that contribute to pathogen resistance. Key components of the immune system—such as major histocompatibility molecules (MHC), immunoglobulins, cytokines, interleukins, T and B cells, and CD4+ and CD8+ T lymphocytes, which are involved in host defense—play critical roles in disease resistance. The breeding of disease-resistant chicken lines can help control pathogens and increase the understanding of host genetics in the control of infectious diseases [124,125]. Advanced technologies, including the CRISPR/Cas9 gene-editing system, whole genome sequencing, RNA sequencing, and high-density SNP genotyping, are instrumental in developing disease-resistant breeds. These approaches have the potential to significantly reduce the use of antibiotics and vaccination in the poultry industry [125]. In general, cellular and humoral immune responses vary among chicken breeds, and high expression of cytokines results in increased host immunocompetence. Different classes of non-coding RNAs, such as circular RNAs, small interfering RNAs (siRNAs), long non-coding RNAs (lncRNAs), miRNAs, and transfer RNAs, also play important roles in avian immunity and regulation of intracellular signaling pathways of cell development, immune response, and oxidative stress [126,127] (Table 2). Several studies have identified a role for circRNAs in avian leukemia virus infection. In addition, differentially expressed miRNAs have been detected in infected organs. miRNAs are involved in T- and B-cell activation and regulation of the Jak-STAT pathway. In contrast to lncRNAs and circRNAs, the expression profile and functional mechanism of miRNAs are well studied in relation to disease resistance in chickens [128]. For example, differentially expressed miRNAs have significant effects on oncogenicity, regulation of MAPK, JaK/STAT, and Wnt pathways, and suppression of chronic myeloid leukemia caused by avian leukemia virus in chickens. Non-coding RNAs regulate disease resistance traits, interact with host and pathogen genes, and aid the control of infectious diseases [125]. Viral diseases lead to increased outbreaks, reduced growth and productivity, and immunosuppression in poultry. GWAS analysis on *Gallus gallus domesticus* has identified genetic markers associated with resistance to diseases such as avian influenza and Marek’s disease and identified key immunity-related genes and variants that increase disease resistance in poultry, which will help in breeding and vaccine development [129]. Sulimova et al. revealed that the population of the Spangled Orloff chicken breed is a source of valuable alleles 357 from the microsatellite loci LEI0258 associated with Marek’s disease resistance. The observed frequency of the genotype 357/357 was 48% [130]. A study by Xu et al. on full genomic genotyping of the Affymetrix^®^ Axiom^®^ HD 600 K genotyping array panel on White Leghorn breed lines resistant and susceptible to Marek’s disease showed that several genes, including *SIK*, *SOX1*, *LIG4*, *SIK1*, and *TNFSF13B*, contribute to immunological characteristics and survival. Based on population differential FST analysis, important genes associated with cell death and apoptosis, including *AKT1*, *API5*, *CDH13*, *CFDP*, and *USP15*, have been identified that may be involved in divergent selection during inbreeding [131]. Lin et al. published the results of a genome-wide association study (GWAS) to identify genes associated with increased adaptability of poultry to environmental stressors, including disease resistance [132]. A study by Smith et al. identified 38 QTLRs on 19 chromosomes of chickens associated with resistance to Marek’s disease. Genomic regions significantly associated with resistance to the virus were mapped, and candidate genes were identified. Different QTLR elements have been shown to have a strong genetic association with resistance [133]. The GWAS identified significant loci and candidate genes associated with resistance to *Salmonella pullorum* in *Gallus gallus domesticus*. Among these two candidate genes, *FBXW7* and *LRBA* were identified as the most promising genes involved in resistance to *S. pullorum*. Two other significant loci and their corresponding genes (*TRAF3* and *gga-mir-489*) were associated with carrier status [134].

## 6. Studies on the Genetics of Climate Adaptability of Chickens

One of the important areas of chicken breeding is to create the most favorable conditions for keeping chickens, in particular when the rearing conditions are as close as possible to natural environments. Under such a rearing system, chickens are exposed to various physical and climatic stresses (cold, heat, and wind), infectious diseases, and social stress. Consequently, birds kept under such conditions should have high adaptive capacity and stress tolerance, which should not be accompanied by increased production cost and risk of diseases [138]. Local chicken breeds carry specific sets of genes responsible for adaptation to harsh environmental conditions and resistance to various diseases. Most investigations in the field of thermal regulation genetics focus on adaptation to hot climates. That is due to the fact that the main areas of developed poultry production (Latin America, China, India, and the Middle East) are concentrated in regions with high average annual temperatures [139,140,141,142,143,144,145]. However, a significant population of chickens is kept in cold climates (Russia, Canada, and Northern Europe). Low temperature is a major environmental factor that can limit animal growth and threaten survival. Despite this, there are some chicken gene pools that have adapted to such harsh conditions through selection and have developed various physiological and biochemical mechanisms of adaptation to the cold environment [146]. Xu et al. in a genome-wide scan of the unique Canadian indigenous breed the Chantecler chicken, which is well adapted to extremely cold conditions, identified two important regions related to fat metabolism and the nervous system during cold adaptation. These regions include the malate dehydrogenase 3 (*ME3*) gene associated with fat production and the *ZNF536* gene encoding a highly conserved zinc finger protein that plays an important role in the development of forebrain neurons involved in social behavior and stress response [147]. Whole-genome sequencing of samples from six local Chinese chicken breeds, as well as 4 chicken breeds from regions with different annual rainfall, temperature, and altitude, revealed a number of candidate genes associated with adaptation to tropical, cold, and arid conditions in local chicken breeds (*CORO2A*, *CTNNA3*, *AGMO*, *GRID2*, *BBOX1*, *COL3A1*, *INSR*, *SOX5*, *MAP2*, and *PLPPR*) [123]. In Russia there are several studies conducted to analyze the resistance of chicken breeds to environmental conditions. Whole genome sequencing of the Russian Snow-White chicken breed revealed a wide pool of mutations unique to this breed, associated with unique characteristics of this breed, such as cold tolerance, resistance to viral diseases, and snow-white color of day-old chicks [148]. A number of candidate genes associated with cold adaptation (*SOX5*, *ME3*, *ZNF536*, *WWP1*, *RIPK2*, *OSGIN2*, *DECR1*, *TPO*, *PPARGC1A*, *BDNF*, and *MSTN*) were identified in a study of Ushanka, Orlovsky Mill Fleur, Russian White, and Cornish White egg breeds using the Chicken 50K_CobbCons SNP chip. The Ushanka and Orlovska mille fleur breeds are indigenous breeds known for their cold tolerance, the Russian White is an egg-type breed selected for cold resistance through artificial selection, and the White Cornish is a meat-type breed [149].

Breeding chickens with adaptive capacity to different climatic conditions is an important task to improve the efficiency of poultry production. A significant amount of chicken genetics studies are focused on the identification of key genes contributing to resistance and adaptability, which could play an important role in the future development of poultry production through the selection of individuals and breeds with high environmental tolerance.

## 7. Prospects for Genetic Improvement of Chickens

Identification of the genetic mechanisms underlying economically valuable traits allows further improvement of chicken breeds through molecular techniques. Germline transmission of desired traits requires modification of individual cells. However, embryo manipulation is challenging in birds, as direct access to one-cell zygotes is difficult. This technique requires different approaches from those successfully used for mammalian species. Nevertheless, genetic engineering and genome editing in chickens have progressed significantly, driven by the need to enhance agricultural traits, improve disease resistance, and develop biomedical applications. Figure 2 demonstrates the methods of genetic engineering and genome editing, including the use of plasmids, viral vectors, and ribonucleoproteins (RNPs).

The first transgenic chicken was produced via the injection of an avian leukosis virus vector into the yolk sac of a laid egg in 1987. Other retroviral vectors, including reticuloendotheliosis virus, and avian sarcoma-leukosis virus, equine infectious anemia virus, were also successfully applied. Although viral injection can reach germline stem cells in the blastoderm, it usually has low efficiency and a high probability of mosaic genotype in the offspring [150]. Later manipulations with the chicken genome were greatly advanced by the use of primordial germ cells (PGCs). These cells are precursors to sperm and oocytes and can be isolated from embryonic blood or gonads, where they can be maintained in long-term culture without losing germline capacity. Genetic modification of cultured PGCs is achieved by standard in vitro transfection or infection techniques. Following selection, modified PGCs can be injected into the recipient embryo through a small opening inside the laid egg to migrate naturally to the developing gonads, mimicking the behavior of endogenous PGCs. Hatched chicks are germline chimeras, which can transmit artificial genetic changes to the offspring, as their gonads contain both transformed and wild-type PGCs [151,152,153]. The cut-and-paste DNA transposon systems piggyBac and Tol2 were efficient in chicken PGCs, allowing the introduction of target genes flanked by inverted terminal repeats into the genome [150].

Sperm-mediated and testis-mediated gene transfer (SMGT and TMGT) are commonly observed in mammals; however, studies in birds are very rare due to the specific sperm structure and chromatin packaging in birds. Using sperm cells as vectors to deliver foreign DNA, typically in the form of a plasmid, into an oocyte enables rapid germline transmission, but efficiency is very low. Washing from nucleases and incubation with DNA greatly reduce the viability of avian sperm. TMGT gives more promising results, as it introduces foreign DNA directly into the testicular tissue, targeting spermatogonial stem cells or developing sperm. Manipulations with the sperm are not required, and transformed chickens can mate naturally or via artificial insemination [154,155]. Chicken testicular cells transformed with a viral vector can also be transplanted to sterilized cockerels, resulting in exogenous spermatogenesis and introduction of genetic changes into the chicken male germ line [156]. Intracytoplasmic sperm injection (ICSI) into an oocyte is a new technique in birds that has not yet resulted in stable transgenic chickens. The method is complicated by natural avian polyspermy and challenges in obtaining unfertilized eggs, visualizing injection, and embryo culture. Hatching a live chick after ICSI currently requires transferring the embryo to a shell or artificial incubation system instead of a hen’s reproductive tract [157]. Chicken embryonic fibroblast cultures are a useful model for viral vector replication, gene function, and assessing the efficiency of CRISPR/Cas9-mediated gene editing [158,159]. However, somatic cell nuclear transfer from transformed fibroblasts is a well-developed technique in mammals; in avian species, it cannot be applied due to oviparous reproduction. To date, researchers have successfully created transgenic chickens with *GFP* and *mCherry* reporter genes and therapeutic genes such as human interferon or Monoclonal antibody genes [160]. The introduction of the *GFP* gene into the Z chromosome allowed to determine the sex of embryos during incubation by fluorescence [161]. However, despite enormous potential, transgenes increasing productivity, disease resistance, or nutritional value were not introduced into chickens. This could be due to regulatory and public acceptance barriers. Nonetheless, the development of genome editing technologies is bringing agricultural applications closer to feasibility. The chicken genome was first edited in 2014 by TALEN [162] and in 2016 by CRISPR/Cas [163]. TALENs are engineered proteins composed of a DNA-binding and FokI nuclease domain enabling them to recognize specific nucleotide sequences and create double-strand breaks (DSBs) in DNA when two TALENs bind adjacent sites on opposite strands. CRISPR/Cas is an RNA-guided system, where Cas nuclease introduces a DSB determined by a single guide RNA (sgRNA). Both systems can be delivered either as performed molecules or encoded within a vector. Once the DSB is induced, the cell’s DNA repair machinery processes the break, introducing mutations. Mutations can arise spontaneously during non-homologous end joining (NHEJ) or be induced precisely through homology-directed repair (HDR) [164,165]. Regulatory approval for genome-edited animals created via Site-Directed Nuclease without template (SDN1) is rather easy in the US, UK, Japan, Brazil, Argentina, and Australia. There is a worldwide trend for deregulation of SDN1 animals, and the list of the countries will continue to grow [166]. To date, at least 8 genes have already been edited in chickens using CRISPR and TALEN technologies [167]. The Ovalbumin gene encoding a major egg white protein, which is non-essential for embryo development, was the first to be edited in order to repurpose the egg white protein locus for biopharmaceutical protein production [162]. The idea developed further, resulting in the knockout of the major allergenic protein ovomucoid without abolishing chicken fertility [168]. Knockout of the Myostatin gene resulted in increased muscle mass and reduced abdominal fat deposition [169]. Editing of the disruption of the G0/G1 switch gene 2 contributed to a dramatic reduction of abdominal fat deposition and altered fatty acid composition [170]. CRISPR/Cas9 deleted a key amino acid (W38) in the avian leukosis virus receptor gene, resulting in the resistance to the infection without any visible side effect [171]. Two precise substitutions (N129I and D130N) in the ANP32A protein significantly reduced chicken susceptibility to the influenza A virus [172]. Genome editing has been used to study the role and functions of several genes in chickens, such as *Stra8* (stimulated by retinoic acid gene 8) and *C2EIP* (Chromosome 1 Expression in PGCs) in cell cultures and embryos [166,173].

Methods of chicken transformation have been optimized to carry CRISPR/Cas elements in RNP or plasmid form. The SMGT approach evolved into STAGE (sperm transfection-assisted gene editing), a technique with the potential to produce genome-edited birds in one generation. Embryos with CRISPR-induced mutations in the GFP and the doublesex and mab-3 related transcription factor 1 (*DMRT1*) genes were successfully generated. Downregulation of *DMRT1* expression leads to gonadal feminization and can be useful in layer hen production [157,165,174].

In summary, chicken genetic engineering and genome editing have advanced rapidly, evolving from early viral transgenesis to the development of precise CRISPR-engineered lines using CRISPR technology. Most germline editing relies on editing PGCs, but other approaches are also available. These technologies can be used to validate the new data on target genes and improve agricultural characteristics of chickens, including resistance to infections and egg and meat production.

## 8. Conclusions

The domestic chicken (*Gallus gallus domesticus*) is a key species in world agriculture, providing mankind with a significant proportion of animal protein and eggs. Advances in next-generation sequencing (NGS) and high-throughput genotyping (SNP chips) technologies have enabled the identification of millions of genetic variants and the mapping of quantitative trait loci underlying economically valuable traits. GWAS (whole-genome association search) studies and QTL analyses have identified genetic markers associated with feed utilization efficiency, growth rate, body weight, meat quality, and fat deposition. The information obtained can be used to improve existing breeds and create new ones. Investigation of the “host-microbiome-environment triad” can be used to assess the impact of different feeds on the microbiome, allowing for the optimization of chicken diets and improved productivity and health in the poultry industry. Integrating metagenomic data on the gut microbiome of chickens will make it possible to complement reference databases and create a catalog for further description of the functional characteristics of various microorganisms. Of critical importance is the further expansion of genomic and post-genomic research using industrial breeds of birds, already in close connection with the objectives of industrial partners in the framework of breeding programs and based on current production tasks.

## Figures and Tables

**Figure 1 ijms-26-08285-f001:**
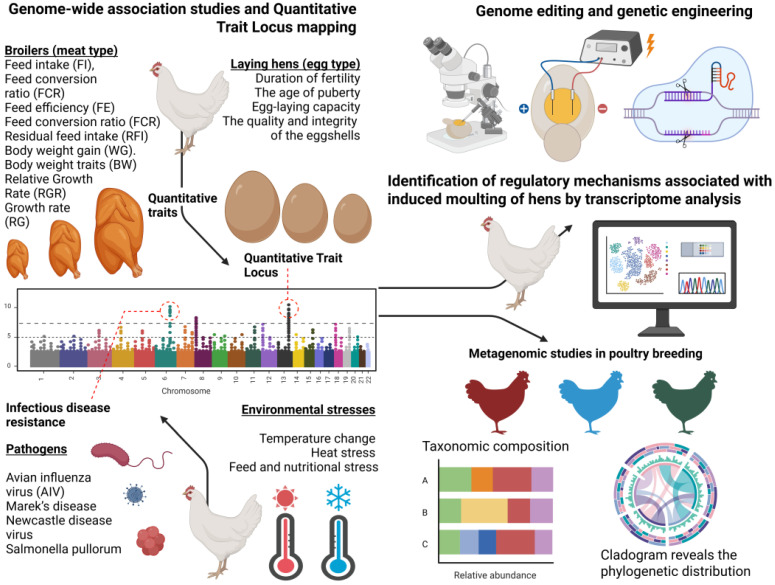
The genomic and post-genomic studies in poultry breeding. (A is the first chicken, B is the second chicken, C is the third chicken).

**Figure 2 ijms-26-08285-f002:**
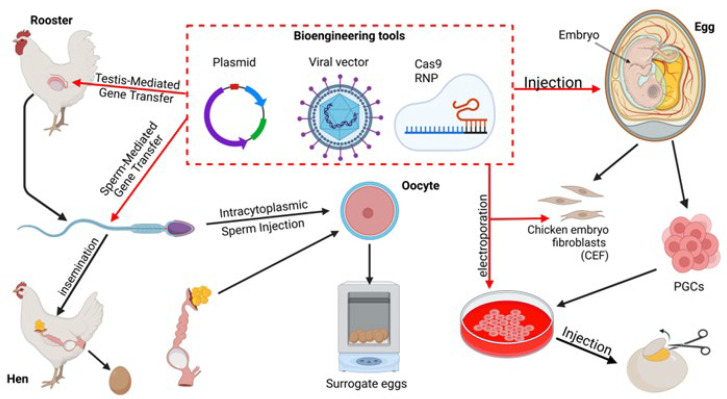
Approaches to genetic improvement of chickens.

**Table 1 ijms-26-08285-t001:** Summary of the genome-wide association and transcriptome analysis studies with chicken breeding traits.

Trait	SNP ID	RefSNP	Gene	Genomic Location	Alleles	*p*-Value	Reference
residual feed intake (RFI)	gga27299173	rs794348453	ENSGALG00000005551 (*DKK3*)	chr5:8134988	T/A	8.62 × 10^−7^	[43]
	gga19252996	rs735238610	ENSGALG00010025920 (*GTSF1*)	chr27:1220239	A/G	1.55 × 10^−7^	[47]
	gga10101246	rs314351418	ENSGALG00000004834 (*COPS3*)	chr14:4782740	A/G	4.53 × 10^−7^	[47]
	gga10101220	rs741733192	ENSGALG00010017495 (*COPS3*)	chr14:4782376	A/G	2.33 × 10^−7^	[47]
feed intake (FI)	gga13915072	rs14175506	ENSGALG00010019808	chr2:45414140	T/C	1.24 × 10^−5^	[43]
	gga40794011	-	-	chr5:6549732	C/T	4.17 × 10^−5^	[48]
	gga27235132	rs315684001	ENSGALG00010024055	chr5:6628773	A/G	9.12 × 10^−5^	[48]
	gga28669902	rs15718055	ENSGALG00010016935	chr5:45362683	C/A	7.24 × 10^−5^	[48]
	gga11900846	rs313913143	ENSGALG00010029371	chr19:1878406	A/C	2.45 × 10^−5^	[48]
	gga19012391	rs15467593	ENSGALG00000038948 (*KDM5B*)	chr26:539371	C/T	3.16 × 10^−5^	[48]
	gga16547347	rs15150566	-	chr2:122907241	A/G	1.1 × 10^−5^	[5]
feed conversion ratio (FCR)	gga26455105	rs738402168	ENSGALG00000041854	chr4:76507176	A/C	9.77 × 10^−42^	[49]
	gga26459075	rs80762609	ENSGALG00010011096	chr4:76588703	A/G	8.52 × 10^−14^	[49]
	gga26453154	rs316119236	ENSGALG00000014421 (*LCORL*)	chr4:76440302	A/T	6.62 × 10^−21^	[49]
	gga26456559	rs740878900	ENSGALG00010010997	chr4:76541002	C/G	6.51 × 10^−21^	[49]
	gga26457701	rs14739738	ENSGALG00000030070 (*QDPR*)	chr4:76563623	A/G	4.24 × 10^−15^	[49]
	gga26420206	rs740848166	ENSGALG00010010298	chr4:75224686	A/T	3.7 × 10^−13^	[49]
	gga26463849	rs80564409	ENSGALG00000014485 (*LDB2*)	chr4:76748107	G/A	3.74 × 10^−16^	[49]
	gga8697511	rs313167401	ENSGALG00010028273	chr12:5990423	T/G	3.74 × 10^−10^	[49]
	gga26455535	rs738469542	ENSGALG00010010979	chr4:76516142	A/G	3.5 × 10^−43^	[49]
	gga26408980	rs316896826	-	chr4:74916879	T/A	3.5 × 10^−19^	[49]
	gga26432816	rs16436155	ENSGALG00000041121 (*SLIT2*)	chr4:75683233	T/G	3.3 × 10^−15^	[49]
	gga26470496	rs316070373	ENSGALG00000014485 (*LDB2*)	chr4:76923106	C/T	2.85 × 10^−13^	[49]
	gga26408556	rs738916134	ENSGALG00010010027	chr4:74884976	C/A	2.34 × 10^−10^	[49]
	gga26456458	-	ENSGALG00000006334 (*LAP3*)	chr4:76538559	C/T	2.21 × 10^−13^	[49]
egg number	gga40794121	-	-	chr5:48969832	A/	1.4 × 10^−18^	[38]
	gga40794155	-	ENSGALG00000037911	chr5:48997447	T/	5.54 × 10^−13^	[38]
	gga40794267	-	ENSGALG00000037911	chr5:49015358	T/	2.48 × 10^−12^	[38]
	gga40794301	-	ENSGALG00000011244 (*DLK1*)	chr5:49016281	G/	5.76 × 10^−12^	[38]
	gga40794078	-	ENSGALG00000042132 (*SUGP1*)	chr28:3540022	T/	1.4 × 10^−10^	[38]
	gga40794259	-	ENSGALG00000037911	chr5:48988619	C/	9.02 × 10^−10^	[38]
	gga40794420	-	ENSGALG00000021598	chr21:5262861	G/	1.55 × 10^−9^	[38]
egg shell thickness	gga19003886	rs736368342	ENSGALG00000000606 (*ARL8A*)	chr26:348743	A/G	3.76 × 10^−7^	[22]
	gga19003889	rs735278795	ENSGALG00000000606 (*ARL8A*)	chr26:348810	G/A	3.76 × 10^−7^	[22]
	gga5865222	rs13968878	ENSGALG00000016967 (*ENOX1*)	chr1:166941530	G/A	2.81 × 10^−8^	[32]
	gga19003886	rs736368342	ENSGALG00000000606 (*ARL8A*)	chr26:348743	A/G	3.76 × 10^−7^	[22]
egg shell weight	gga15205102	rs13636444	ENSGALG00010008389 (*GALNT1*)	chr2:84108965	G/A	5.85 × 10^−9^	[32]
	gga23503286	rs14411624	ENSGALG00010011743	chr3:107750850	C/T	1.41 × 10^−7^	[32]
	gga8069820	rs14022717	ENSGALG00010024544	chr11:9050275	G/T, A	8.62 × 10^−7^	[32]
age at first egg	gga9401651	rs318240317	ENSGALG00000001768 (*TENM2*)	chr13:5042404	T/C	1.42 × 10^−6^	[32]
egg shell percentage	gga11229394	rs793955278	ENSGALG00010027632	chr17:6527642	C/A	2.98 × 10^−7^	[22]
	gga14284065	rs317955040	-	chr2:57645767	C/T	5.53 × 10^−10^	[22]
	gga20208557	rs793960563	ENSGALT00010040903 (*SLC8A1*)	chr3:16388607	A/G	1.52 × 10^−9^	[22]
	gga20618399	rs15305641	-	chr3:27338039	G/A	1.43 × 10^−8^	[22]
	gga2247149	rs14834812	ENSGALG00000013037 (*BCL2L13*)	chr1:61905467	T/C	1.74 × 10^−7^	[22]
	gga7655209	rs740613354	ENSGALG00010018848 (*MEF2A*)	chr10:17024031	T/G	5.56 × 10^−8^	[22]
body weight at first egg	gga26454149	rs16023603	ENSGALG00000041854	chr4:76488977	G/T	9.75 × 10^−8^	[50]
egg shell color	gga10971513	rs731126327	ENSGALG00000037735 (*CENPA*)	chr16:58557	C/G	2.57 × 10^−9^	[22]
	gga18068242	rs315477097	ENSGALP00000043161	chr21:803620	G/C	3.83 × 10^−8^	[51]
	gga17077724	rs15168063	ENSGALP00000045123	chr2:137478073	T/A	4.63 × 10^−8^	[22]
	gga11908141	rs14117102	ENSGALG00010029870	chr19:2048452	T/C	2.01 × 10^−8^	[22]
	gga36013348	rs793971423	ENSGALG00010009886	chrZ:73194176	C/T	8.98 × 10^−9^	[22]
	gga6512746	rs315232554	ENSGALP00000051533	chr1:183114617	T/C	1.5 × 10^−9^	[22]
	gga18068999	rs313199923	ENSGALG00010019630	chr21:822838	T/C	3.29 × 10^−6^	[51]
	gga13707557	rs316634461	ENSGALP00000030515	chr2:40713376	G/A	2.77 × 10^−6^	[51]
	gga18072343	rs16177221	ENSGALG00000000978 (*CEP104*)	chr21:912580	G/A	2.84 × 10^−6^	[51]
breast muscle weight	gga19367973	-	ENSGALG00000041204 (*IGF2BP1*)	chr27:3929034	A/C	3.09 × 10^−8^	[52]
	gga19367739	rs741713216	ENSGALG00000041204 (*IGF2BP1*)	chr27:3923534	T/C	1.5 × 10^−8^	[52]
	gga19360817	rs733674119	ENSGALG00000001315 (*UBE2Z*)	chr27:3696784	T/C	1.89 × 10^−8^	[52]
	gga19361902	rs740150938	ENSGALG00000041204 (*IGF2BP1*)	chr27:3727891	A/T	1.38 × 10^−8^	[52]
	gga19366244	rs739298135	ENSGALG00000041204 (*IGF2BP1*)	chr27:3883310	A/G	6.21 × 10^−9^	[52]
	gga19360904	rs14303799	ENSGALG00000041204 (*IGF2BP1*)	chr27:3699719	T/C	5.17 × 10^−9^	[52]
	gga19361431	rs737533546	ENSGALG00000001525 (*CALCOCO2*)	chr27:3713052	C/T	4.16 × 10^−9^	[52]
	gga19361079	rs736156149	ENSGALG00000001330 (*ATP5G1*)	chr27:3705603	G/A	2.39 × 10^−9^	[52]
	gga26142808	rs313870616	ENSGALG00010017374	chr4:66917344	T/C	2.79 × 10^−11^	[53]
body weight	gga2548108	rs315749637	-	chr1:69658667	G/A	9.86 × 10^−9^	[53]
	gga26144149	rs315209025	ENSGALG00000014124 (*TEC*)	chr4:66946846	C/A	1.38 × 10^−10^	[53]
	gga26144434	rs315209025	ENSGALG00000014124 (*TEC*)	chr4:66952728	C/A	3.67 × 10^−12^	[53]
	gga32875195	rs313957679	ENSGALG00000010543 (*EPS15*)	chr8:24178278	G/A	6.09 × 10^−9^	[53]
	gga33041294	rs14657331	ENSGALG00010024151 (*LEPROT*)	chr8:28421453	C/T	2.64 × 10^−14^	[53]
	gga33080862	rs312436211	ENSGALG00000011350 (*NEGR1*)	chr8:29394332	C/T	9.16 × 10^−11^	[53]
	gga10122855	rs14073523	ENSGALG00000005215 (*CACNA1H*)	chr14:5337950	A/G	3.8 × 10^−5^	[48]
body weight gain	gga11364204	rs312843163	ENSGALG00000001094 (*ADGRD2*)	chr17:9904101	A/G	5.13 × 10^−6^	[48]
	gga16591237	rs15151359	-	chr2:124071457	G/A	3.72 × 10^−5^	[48]
	gga28701462	rs316866456	ENSGALG00010018389	chr5:46165348	C/T	8.13 × 10^−5^	[48]
	gga26455105	rs738402168	ENSGALG00000041854	chr4:76507176	A/C	1.39 × 10^−62^	[49]
average daily gain	gga26454915	rs315397301	ENSGALG00000041854	chr4:76503015	T/C	2.21 × 10^−60^	[49]
	gga26408556	rs738916134	ENSGALG00000040208	chr4:74884976	C/A	4.86 × 10^−31^	[49]
	gga26450824	rs80691090	-	chr4:76367855	G/C	2.75 × 10^−29^	[49]
	gga26459318	rs16756269	ENSGALG00010011096	chr4:76593451	G/A	2.98 × 10^−29^	[49]
	gga26455535	rs738469542	ENSGALG00010010979	chr4:76516142	A/G	5.11 × 10^−28^	[49]
	gga26430969	rs315846457	ENSGALG00000041121 (*SLIT2*)	chr4:75622029	A/G	1.25 × 10^−25^	[49]
	gga11356458	rs316227600	ENSGALG00000001157 (*DENND1A*)	chr17:9636765	T/C	2.21 × 10^−6^	[54]

**Table 2 ijms-26-08285-t002:** Summary of the genome-wide association studies with chicken traits related to resistance against infectious diseases.

Trait	SNP ID	RefSNP	Gene	Genomic Location	Alleles	*p*-Value	Reference
salmonella enterica serovan gallinarum antibody titre	gga11356458	rs316227600	ENSGALG00000001157 (*DENND1A*)	chr17:9636765	T/C	2.21 × 10^−6^	[54]
	gga14606249	rs313247175	ENSGALG00000027339 (*LYRM4*)	chr2:65707454	T/C	5.9 × 10^−5^	[135]
	gga11356599	-	ENSGALG00000001157	chr17:9642868-9642868	A/C	1.21 × 10^−6^	[54]
avian influenza virus antibody titer	gga29091675	rs14554319	ENSGALG00000012137 (*KTN1*)	chr5:56259597	T/C	5.44 × 10^−5^	[135]
	gga28738351	rs16505398	-	chr5:47167537	C/T	2.69 × 10^−14^	[136]
antibody level against infectious bronchitis virus	gga4586819	rs13623466	ENSGALG00000016681 (*DHRSX*)	chr1:128713658	C/T	9.98 × 10^−14^	[136]
	gga32379046	rs314472262	ENSGALG00000004620 (*LAMC1*)	chr8:7544464	T/C	1.8 × 10^−13^	[136]
	gga4604674	rs313566132	ENSGALG00000016689 (*ASMTL*)	chr1:129172183	C/T	2.19 × 10^−13^	[136]
	gga11638399	rs14112036	ENSGALG00000003105 (*ANKFN1*)	chr18:6199395	T/C	7.22 × 10^−6^	[54]
mareks disease virus antibody titre	gga21222152	rs14346868	ENSGALG00000011473 (*RPS6KA2*)	chr3:42937011	T/C	1.05 × 10^−6^	[54]
	gga18354446	rs15998498	ENSGALG00000001608 (*UNC5D*)	chr22:1854894	C/T	6.75 × 10^−6^	[137]
pre-infection growth rate	gga21954944	rs317939411	ENSGALG00000014902	chr3:63366122	C/T	5.42 × 10^−6^	[137]
post-infection growth rate	gga11889533	rs314290710	ENSGALG00000001153 (*AUTS2*)	chr19:1607256	G/T	3.65 × 10^−6^	[137]
immune response to newcastle disease	gga1757454	rs314284996	ENSGALG00000011894 (F1NJG4)	chr1:49441152	T/C	1.55 × 10^−7^	PMID 34745207
	gga1822646	rs737774287	ENSGALG00000012291 (POLR2F)	chr1:51104995	T/C	4.0 × 10^−10^	PMID 34745207
	gga40793944	-	ENSGALG00000041823	chr1:51056044	A/C	6.42 × 10^−8^	PMID 34745207
resistance to cestodes parasitism	gga31494288	-	ENSGALG00000011318.8 (DNPEP)	chr7:22288125	T/C	1.8 × 10^−11^	[54]
	gga10144957	-	ENSGALG00000044187 (LMF1)	chr14:5838224	T/C	1.06 × 10^−8^	[54]
resistance to eimeria parasitism	gga21878395	-	ENSGALG00000014848 (TRDN)	chr3:61008720	C/T	8.09 × 10^−7^	[54]
	gga40793947	-	ENSGALG00000030397	chr16:161178-161178	C/A	1.0 × 10^−5^	[54]

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
