# Peer review of "Advances in Genomics and Postgenomics in Poultry Science: Current Achievements and Future Directions"

_ijms, 2025, doi:10.3390/ijms26178285_

Round 1
Reviewer 1 Report
Comments and Suggestions for Authors
The review paper titled "Advances in Genomics and Postgenomics in Poultry Science: Current Achievements and Future Directions" provides a comprehensive overview of recent advances in chicken genomic and transcriptomic research, particularly those focused on improving meat production traits and reproductive performance. Notably, the paper effectively integrates multi-omics approaches (e.g., genome-wide association studies, metagenome-assembled genomes) with industrial breeding objectives, highlighting their synergistic potential for accelerating genetic selection. The systematic organization of content, from foundational genomic techniques to cutting-edge microbiome interactions, demonstrates thorough scholarship in this evolving field. The paper is well-structured and clearly written, with high-quality English throughout and precise visualizations of key datasets. While the technical depth is commendable, one suggestion for improvement is that the conclusion could be made shorter and more concise by synthesizing proposed research priorities (e.g., host-microbiome-environment triad analysis) into actionable frameworks for both academia and industry. Some minor revisions have been marked in the PDF version. These are offered as optional suggestions for the authors to consider if they find them appropriate, with particular attention to enhancing translational relevance for commercial poultry operations.

Reviewer 2 Report
Comments and Suggestions for Authors
A review manuscript that synthesizes advances in genomics and post-genomics applied to poultry science, focusing on genomic studies (GWAS/QTL), transcriptomics, intestinal microbiome metagenomics, and genome-editing tools (CRISPR/Cas). The text covers the discovery of variants, candidate genes for productive and reproductive traits, disease resistance, and aspects of egg and meat quality.
Points for improvement
1 - Describe the strategy used to include the literature covered and the period the review encompasses (last 10 years?).
2 - Line 366 – The text indicates that the data are contradictory, but does not specify what the contradictions are. This must be made clearer for the reader.
3 - In several places the manuscript points out and lists identified genes and QTLs, but in most cases does not discuss the statistical approaches used or the level of significance.
Minor corrections
Line 377 – The authors state that “There is an evidence of the significant impact of the chicken microbiome on egg qual-377 ity and safety” but do not describe what that evidence is nor the strength of the evidence.
Major corrections
Include more critical paragraphs that explain what the divergences between studies are (methodological limits, statistical power, and population differences).
Reviewer 3 Report
Comments and Suggestions for Authors
This is a much-needed and timely review in the field of genomics within poultry science. The manuscript is well written and effectively structured, covering recent advances in both genomics and post-genomics with clarity and depth. I have minor suggestions for improvement. In Section 5, where the manuscript discusses genetic studies related to resistance against infectious diseases, the inclusion of a table listing specific resistance genes along with their associated pathogens (bacterial or viral) would enhance readability and utility for readers. Additionally, Section 6, which addresses the genetics of climate adaptability, would benefit from a concluding paragraph summarizing the key insights and implications of the studies presented. Aside from these small suggestions, the article is well designed and represents a valuable contribution to the literature.
